# Learning Data for Neural-Network-Based Numerical Solution of PDEs: Application to Dirichlet-to-Neumann Problems



**Ferenc Izsák** [1,2,*,†] and **Taki Eddine Djebbar** [2,†]

1   Alfréd Rényi Institute of Mathematics, 1053 Budapest, Hungary
2   Department of Applied Analysis, Eötvös Loránd University, 1117 Budapest, Hungary
*   Correspondence: ferenc.izsak@ttk.elte.hu
†   These authors contributed equally to this work.

**Abstract:** We propose neural-network-based algorithms for the numerical solution of boundary-value problems for the Laplace equation. Such a numerical solution is inherently mesh-free, and in the approximation process, stochastic algorithms are employed. The chief challenge in the solution framework is to generate appropriate learning data in the absence of the solution. Our main idea was to use fundamental solutions for this purpose and make a link with the so-called method of fundamental solutions. In this way, beyond the classical boundary-value problems, Dirichlet-to-Neumann operators can also be approximated. This problem was investigated in detail. Moreover, for this complex problem, low-rank approximations were constructed. Such efficient solution algorithms can serve as a basis for computational electrical impedance tomography.

**Keywords:** boundary-value problems; neural networks; Dirichlet-to-Neumann problem; learning data; method of fundamental solutions

## 1. Introduction

Using artificial neural networks (ANNs), a number of computationally complex or data-intensive computational problems have been successfully tackled in the last decade. Furthermore, ANNs can be applied to computational tasks with no rigorous mathematical model in the background; the structure of the neural network itself serves as a (sometimes, heuristic) model.

A natural idea is to employ this powerful tool for the classical computational problems such as the numerical solution of partial differential equations (PDEs). Accordingly, a number of different algorithms have been proposed.

The most-widespread family of the different approaches is the so-called physics-informed neural networks (PINNs), which were introduced in [1]. For a detailed review on this topic, we refer to [2].

In the first version of this popular family of methods, called the data-driven PINNs, we have a fixed PDE $\partial_t u(t, \mathbf{x}) + D_{\mathbf{x}} u(t, \mathbf{x}) = 0$ for the unknown function $u : (0, T) \times \Omega \to \mathbb{R}$, where $\Omega$ denotes the spatial computational domain and $D_{\mathbf{x}}$ is a generic spatial differential operator. Furthermore, the equation should be equipped with an appropriate initial condition $u_0$ and boundary data $u_{\partial \Omega}$ for the well-posedness.

To prepare the computations with the ANN, a huge number of solution candidates $v_j$ should be computed such that these determine a large variety of initial and boundary conditions. If $v_j \approx u$ is only an approximation, the right-hand side defined by $f_j := \partial_t v_j + D_x v_j$ is not necessarily zero. Furthermore, $v_j$ has some initial value $v_{j,0}$ and boundary value $v_{j,\partial\Omega}$. Within the frame of the conventional PINNs, for a given set $\{\mathbf{x}_j\}_{j=1}^M \subset \Omega$, an ANN is trained using the pairs:

$$\left\{ \left( [f_j, v_{j,0}, v_{j,\partial\Omega}], [v_j(t_1, \mathbf{x}_1), \dots, v_j(t_M, \mathbf{x}_M)] \right) \right\}. \tag{1}$$

In this way, the ANN will try to learn the solution at some points using the left-hand side $f_j$, the boundary data $v_{j,\partial\Omega}$, and the initial data $v_{j,0}$, such that it can be recognized as a solution operator. For a given structure of the ANN, the values of its parameters will be determined such that the error:

$$[v_j(t_1, \mathbf{x}_1), \ldots, v_j(t_M, \mathbf{x}_M)] - [\hat{v}_j(t_1, \mathbf{x}_1), \ldots, \hat{v}_j(t_M, \mathbf{x}_M)]$$

is minimal. Here, $\hat{v}_j$ denotes the ANN estimate of the original function $v_j$. To perform this, we should specify a norm to measure the error above. This norm is commonly called the loss function. Finally, applying this network to the input $(0, u_0, u_{\partial\Omega})$, we obtain the desired approximation of $u$.

This original idea was developed and extended to incorporate unknown parameters in the equation. In this way, it is perfectly fit to solve inverse problems; see, e.g., [3]. The real-life case of noisy measurements or observation data can also be fit into this framework, linking this computational tool with the powerful Bayesian inversion approach [4].

Furthermore, the methodology of computing with PINNs was refined and discussed for a number of cases concerning the Navier–Stokes equations [5], as well as stochastic PDEs [6]. Beyond these specific applications, the PINN-based methods were also extended to make them flexible in the choice of the computational domain [7].

Due to the practical importance of developing efficient PDE solvers, a number of further ANN-based approaches have been proposed for this purpose. A new, promising methodology is based on mappings between function spaces. As the Galerkin methods became the leading tool in the case of conventional numerical methods, the function space approach, instead of working pointwise or with mesh-dependent data, seems to be an adequate direction for the learning algorithms. This was described in [8], where also an extensive review is provided on further ANN-based methodologies.

Our present study was restricted to the numerical solution of the Laplace equation and, in particular, the corresponding Dirichlet-to-Neumann problem. At the same time, we combined here the knowledge of the classical analysis with the powerful tools given by the neural networks.

The numerical solution of the Laplace equation is a rather classical topic: It serves as an important building block for practical problems such as non-linear water waves [9] or electrical impedance tomography (EIT), where very flexible and fast solvers for elliptic-boundary-value problems are needed. In the corresponding mathematical model, the Dirichlet-to-Neumann map is known at some boundary points. Using this information, the parameters of the original elliptic differential operator should be determined.

As EIT is an important non-invasive diagnostic tool in medicine, the corresponding computational algorithms are of interest [10]. Accordingly, ANN-based approaches have also been elaborated for this; see [11,12].

Furthermore, the methodology we develop here can be easily extended to a number of PDEs, where the fundamental solution is known.

After stating the computational problem, we discuss some necessary tools in the mathematical analysis. Then, the main principles of the proposed algorithm are described in detail with the link to the classical methods of the numerical analysis. Afterwards, we give the algorithm in concrete terms. In the final section, the details of the implementation are given, and the efficiency of the present approach is demonstrated by some numerical experiments.

## 2. Materials and Methods

We first state the mathematical problems to be solved.

### 2.1. Problem Statement and Mathematical Background

The focus of the present study is the ANN-based approximation of some mathematical problems based on the Laplacian equation:

$$\begin{cases} \Delta u(\mathbf{x}) = 0 & \mathbf{x} \in \Omega \\ u(\mathbf{x}) = g(\mathbf{x}) & \mathbf{x} \in \partial\Omega, \end{cases} \tag{2}$$

where $\Omega \subset \mathbb{R}^d$ denotes the computational domain, $u : \Omega \to \mathbb{R}$ is an unknown function, and $g : \partial\Omega \to \mathbb{R}$ is given.

The corresponding theory (see, e.g., [13]) ensures that this problem is well-posed: for any $g \in H^{\frac{1}{2}}(\partial\Omega)$, it has a unique solution $u \in H^1(\Omega)$ depending continuously on $g$.

Here, it was assumed that $\Omega$ is bounded and has a Lipschitz boundary, and we used the classical Sobolev spaces:

$$H^1(\Omega) = \{u \in L_2(\Omega) : \nabla u \in [L_2(\Omega)]^d\} \quad \text{with} \quad \|u\|_{H^1(\Omega)} = \left( \|u\|_{L_2(\Omega)}^2 + \|\nabla u\|_{[L_2(\Omega)]^d}^2 \right)^{\frac{1}{2}}$$

and

$$H^{\frac{1}{2}}(\partial\Omega) = \{u|_{\partial\Omega} : u \in H^1(\Omega)\} \quad \text{with} \quad \|u\|_{H^{\frac{1}{2}}(\partial\Omega)} = \inf\{\|\tilde{u}\|_{H^1(\Omega)} : \tilde{u}|_{\partial\Omega} = u\}.$$

In practice, however, $g$ is not given in all points of $\partial\Omega$, and possibly, $u$ is sought only in some interior points. Accordingly, as an introductory problem, we discuss the following case:

**Problem 1.** *For given values of* $g(\mathbf{z}_1), g(\mathbf{z}_2), \ldots, g(\mathbf{z}_N)$ *with* $\{\mathbf{z}_j\}_{j=1}^N \subset \partial\Omega$, *we should determine* $u(\mathbf{x}_1), u(\mathbf{x}_2), \ldots, u(\mathbf{x}_M)$ *with* $\{\mathbf{x}_j\}_{j=1}^M \subset \Omega$.

Our objective was to reduce the computational costs and approximate $u(\mathbf{x}_1), u(\mathbf{x}_2), \ldots, u(\mathbf{x}_M)$ without computing a "complete" numerical solution $u_h \approx u$ of (2).

A related problem is computing the corresponding Dirichlet-to-Neumann map.

**Problem 2.** *For given values of* $g(\mathbf{z}_1), g(\mathbf{z}_2), \ldots, g(\mathbf{z}_N)$ *with* $\{\mathbf{z}_j\}_{j=1}^N \subset \partial\Omega$, *we should determine* $\partial_\nu u(\mathbf{z}_1^*), \partial_\nu u(\mathbf{z}_2^*), \ldots, \partial_\nu u(\mathbf{z}_M^*)$ *with* $\{\mathbf{z}_j^*\}_{j=1}^M \subset \partial\Omega$.

The associated "full" mapping corresponding to (2) is defined as

$$\text{DtN} : H^{\frac{1}{2}}(\partial\Omega) \to H^{-\frac{1}{2}}(\partial\Omega), \ \text{DtN}(g) = \partial_\nu u,$$

where $u \in H^1(\Omega)$ is the solution of (2), $\nu$ denotes the (space-dependent) outward normal on $\partial\Omega$, and $H^{-\frac{1}{2}}(\partial\Omega)$ denotes the dual-space of $H^{\frac{1}{2}}(\partial\Omega)$. This is called the Dirichlet-to-Neumann map. Of course, to develop an efficient procedure, we intended to perform it without approximating $u$ in the entire domain.

In a related mathematical model, the function $g$ corresponds to an applied potential (using electrodes) at the surface of some organ and $\partial_\nu u$ corresponds to the generated current on the boundary. Here, instead of the Laplacian, we used a general elliptic operator $\nabla \cdot (\mu \nabla)$, where the space-dependent function $\mu : \Omega \to \mathbb{R}$ describes the permittivity.

In this way, the Dirichlet-to-Neumann map can be measured, and in a real-life inverse problem (in EIT), the corresponding permittivity has to be determined.

**Remark 1.** *Note that neither Problem 1 nor Problem 2 are well-posed. Therefore, we cannot perform a conventional error analysis.*

Furthermore, an objective of our approach was to include and use knowledge from the PDE theory and the conventional numerical methods in the ANN-based algorithm. The most-important tool is the use of the fundamental solution, which, for $d = 2$, is defined as

$$\Psi : \mathbb{R}^2 \setminus \{\mathbf{0}\} \to \mathbb{R}, \ \Psi(\mathbf{x}) = -\ln|\mathbf{x}|.$$

For a given $\mathbf{y} \in \mathbb{R}^2$, we also used its shifted version, which is given by $\Psi_{\mathbf{y}}(\mathbf{x}) = -\ln|\mathbf{x} - \mathbf{y}|$.

As the main objective, we intend to construct neural networks, where the vector $g(\mathbf{z}_1), g(\mathbf{z}_2), \ldots, g(\mathbf{z}_N)$ serves as an input according to the given boundary data in (2), and the output will approximate:

- $u(\mathbf{x}_1), u(\mathbf{x}_2), \ldots, u(\mathbf{x}_M)$ in the case of Problem 1;
- $\partial_\nu u(\mathbf{z}_1^*), \partial_\nu u(\mathbf{z}_2^*), \ldots, \partial_\nu u(\mathbf{z}_M^*)$ in the case of Problem 2.

### 2.2. Principles and Details of the Algorithm

Observe that, for $\mathbf{y} \notin \Omega$, we have $\Delta \Psi_{\mathbf{y}}(\mathbf{x}) = 0$ for $\mathbf{x} \in \Omega$, such that it satisfies the first equation in (2). In this way, for a fixed value of $\mathbf{y} \notin \overline{\Omega}$, we used the following.

Main Principle.

The pair:

$$([\Psi_{\mathbf{y}}(\mathbf{z}_1), \Psi_{\mathbf{y}}(\mathbf{z}_2), \ldots, \Psi_{\mathbf{y}}(\mathbf{z}_N)], [\Psi_{\mathbf{y}}(\mathbf{x}_1), \Psi_{\mathbf{y}}(\mathbf{x}_2), \ldots, \Psi_{\mathbf{y}}(\mathbf{x}_M)]) \qquad (3)$$

delivers a learning data pair (input and output) for Problem 1.

Likewise, the pair $([\Psi_{\mathbf{y}}(\mathbf{z}_1), \Psi_{\mathbf{y}}(\mathbf{z}_2), \ldots, \Psi_{\mathbf{y}}(\mathbf{z}_N)], [\partial_\nu \Psi_{\mathbf{y}}(\mathbf{z}_1^*), \partial_\nu \Psi_{\mathbf{y}}(\mathbf{z}_2^*), \ldots, \partial_\nu \Psi_{\mathbf{y}}(\mathbf{z}_M^*)])$ delivers a learning data pair for Problem 2.

Summarizing, the ANN can be trained with the above data for an arbitrary set of points $\{\mathbf{y}_j\}_{j=1}^K \subset \mathbb{R}^2 \setminus \overline{\Omega}$. In this way, we easily obtain a learning dataset consisting of $K$ input–output pairs.

**Remark 2.**

1.  *In practice, we did not choose $\mathbf{y}_j \in \overline{\Omega}$, since in this case, $\Psi$ will have a singularity here, which results in huge values $\Psi_{\mathbf{y}}(\mathbf{z}_j)$ in the vicinity of the boundary.*
2.  *The present approach (3), compared to the conventional one (1), defines a much easier procedure:*
    - *We can get rid of the first term in (1) automatically;*
    - *Furthermore, for a stationary problem, we do not need the initial condition in the second term in (1).*

    *In short, our approach results in a significant reduction of the parameters in the associated ANN. It seems obvious to employ this idea in general, for all stationary boundary values problems, if we know solutions $\Psi_j$ with $D_{\mathbf{x}} \Psi_j = 0$. However, one should be careful here: we can only approximate all possible solutions if we can state some approximation property on the linear hull of the family of functions $\{\Psi_j\}$.*
3.  *In other words, our approach can also be considered as a shortcut in the PINNs. As we do not have to compute solution candidates with different right-hand sides $f_j$, the number of learning data is also reduced, and the ANN can be trained much faster.*

Recall that, in the Dirichlet-to-Neumann problem, the natural norm of the Neumann data is $\| \cdot \|_{H^{-\frac{1}{2}}(\partial\Omega)}$, which can lead to the slow convergence of the result. In the ANN-based algorithm, this can also lead to the slowing down of the optimization procedure.

To try an alternative approach for Problem 2, observe that the Neumann boundary data of $u$ at $\mathbf{z}_j \in \partial\Omega$ can also be estimated using the approximation:

$$\partial_\nu u(\mathbf{z}_j^*) \approx \frac{u(\mathbf{z}_j^*) - u(\mathbf{z}_j^* - s\nu)}{s} \qquad (4)$$

for a small value $s \in \mathbb{R}^+$. Then, we have to estimate $u(\mathbf{z}_j^* - s\boldsymbol{\nu})$ and $u(\mathbf{z}_j^*)$ to obtain the pointwise approximation in (4). A motivation for this approach is the good performance of the estimation of $u(\mathbf{z}_j^* - s\boldsymbol{\nu})$, which is shown later.

*2.3. A Related Classical Approach*

Both Problem 1 and Problem 2 are linear, and accordingly, we intended to construct the ANN as a linear map. In this case, it can be identified with a matrix $A \in \mathbb{R}^{N \times M}$ such that we have the approximation:

$$[\Psi_{\mathbf{y}_j}(\mathbf{x}_1), \Psi_{\mathbf{y}_j}(\mathbf{x}_2), \ldots, \Psi_{\mathbf{y}_j}(\mathbf{x}_M)]^T \approx A[\Psi_{\mathbf{y}_j}(\mathbf{z}_1), \Psi_{\mathbf{y}_j}(\mathbf{z}_2), \ldots, \Psi_{\mathbf{y}_j}(\mathbf{z}_N)]^T. \tag{5}$$

Keeping this in mind, any input vector:

$$[u(\mathbf{z}_1), u(\mathbf{z}_2), \ldots, u(\mathbf{z}_N)]^T = [g(\mathbf{z}_1), g(\mathbf{z}_2), \ldots, g(\mathbf{z}_N)]^T$$

in the ANN as the boundary data is first approximated as

$$[g(\mathbf{z}_1), g(\mathbf{z}_2), \ldots, g(\mathbf{z}_N)]^T \approx \sum_{j=1}^{K} b_j[\Psi_{\mathbf{y}_j}(\mathbf{z}_1), \Psi_{\mathbf{y}_j}(\mathbf{z}_2), \ldots, \Psi_{\mathbf{y}_j}(\mathbf{z}_N)]^T, \tag{6}$$

where $\{b_j\}_{j=1}^K$ are unknown real coefficients. Indeed, taking $K \geq N$ and appropriate values of $\{\mathbf{y}_j\}_{j=1}^K$ such that

$$\text{rank}\left\{[\Psi_{\mathbf{y}_j}(\mathbf{z}_1), \Psi_{\mathbf{y}_j}(\mathbf{z}_2), \ldots, \Psi_{\mathbf{y}_j}(\mathbf{z}_N)]^T\right\}_{j=1}^K = N,$$

the approximation in (6) becomes an equality.

This is the case in real-life situations, since we try to produce a large number $K$ of learning data pairs. Moreover, using the singular property of $\Psi$, for each $\mathbf{z}_k$, we can choose $\mathbf{y}_k$ in its vicinity such that

$$|\Psi_{\mathbf{y}_k}(\mathbf{z}_k)| \geq \sum_{j \neq k} |\Psi_{\mathbf{y}_k}(\mathbf{z}_j)|.$$

This means that the matrix composed by the vectors:

$$\left\{[\Psi_{\mathbf{y}_j}(\mathbf{z}_1), \Psi_{\mathbf{y}_j}(\mathbf{z}_2), \ldots, \Psi_{\mathbf{y}_j}(\mathbf{z}_N)]^T\right\}_{j=1}^N$$

will be diagonally dominant, such that it is non-singular. Then, the rank of its columns is indeed $N$.

In this case, using (5) and (6) with equality, the neural network with the input $[g(\mathbf{z}_1), g(\mathbf{z}_2), \ldots, g(\mathbf{z}_N)]^T$ gives

$$\sum_{j=1}^{K} b_j[\Psi_{\mathbf{y}_j}(\mathbf{x}_1), \Psi_{\mathbf{y}_j}(\mathbf{x}_2), \ldots, \Psi_{\mathbf{y}_j}(\mathbf{x}_M)]^T \approx \sum_{j=1}^{K} b_j A[\Psi_{\mathbf{y}_j}(\mathbf{z}_1), \Psi_{\mathbf{y}_j}(\mathbf{z}_2), \ldots, \Psi_{\mathbf{y}_j}(\mathbf{z}_N)]^T,$$

where the left-hand side is the prediction of $u$ at $\mathbf{x}_1, \mathbf{x}_2, \ldots, \mathbf{x}_M$. In summary, the linear ANN-based approximation can be recognized as an approximation:

$$u \approx \sum_{j=1}^{K} b_j \Psi_{\mathbf{y}_j}.$$

This is exactly the idea of the *method of fundamental solutions* (MFS), which was initiated in [14]. Due to its simplicity and meshless property, this numerical method became extremely popular in the engineering community. At the same time, a rigorous and general convergence theory for the MFS is still missing. In any case, such a theory is based on the

approximation property of the functions $\{\Psi_{\mathbf{y}_j}\}_{j=1}^{K}$. A nice summary of the related results can be found in [15].

Furthermore, the approximation property of our neural network depends on the density properties of the above function space.

*2.4. The Algorithm*

Based on the above approach, we utilized the following algorithm:

1. We define points $\{\mathbf{y}_j\}_{j=1}^{K} \subset \mathbb{R}^2 \setminus \overline{\Omega}$, which are near $\partial\Omega$ and are distributed quasi-uniformly.

2. For Problem 1, we computed the learning input–output pairs

$$\left\{ \left( [\Psi_{\mathbf{y}_j}(\mathbf{z}_1), \Psi_{\mathbf{y}_j}(\mathbf{z}_2), \ldots, \Psi_{\mathbf{y}_j}(\mathbf{z}_N)], [\Psi_{\mathbf{y}_j}(\mathbf{x}_1), \Psi_{\mathbf{y}_j}(\mathbf{x}_2), \ldots, \Psi_{\mathbf{y}_j}(\mathbf{x}_M)] \right) \right\}_{j=1}^{K}.$$

3. For Problem 2, in the first approach, we computed the learning input–output pairs

$$\left\{ \left( [\Psi_{\mathbf{y}_j}(\mathbf{z}_1), \Psi_{\mathbf{y}_j}(\mathbf{z}_2), \ldots, \Psi_{\mathbf{y}_j}(\mathbf{z}_N)], [\partial_\nu \Psi_{\mathbf{y}_j}(\mathbf{z}_1^*), \partial_\nu \Psi_{\mathbf{y}_j}(\mathbf{z}_2^*), \ldots, \partial_\nu \Psi_{\mathbf{y}_j}(\mathbf{z}_M^*)] \right) \right\}_{j=1}^{K}.$$

For Problem 2, in the second approach, we computed first the learning input–output pairs

$$\left\{ \left( [\Psi_{\mathbf{y}_j}(\mathbf{z}_1), \Psi_{\mathbf{y}_j}(\mathbf{z}_2), \ldots, \Psi_{\mathbf{y}_j}(\mathbf{z}_N)], [\Psi_{\mathbf{y}_j}(\mathbf{z}_1^*), \Psi_{\mathbf{y}_j}(\mathbf{z}_2^*), \ldots, \Psi_{\mathbf{y}_j}(\mathbf{z}_M^*)] \right) \right\}_{j=1}^{K}$$

and, then, additionally, the learning input–output pairs

$$\left\{ \left( [\Psi_{\mathbf{y}_j}(\mathbf{z}_1), \Psi_{\mathbf{y}_j}(\mathbf{z}_2), \ldots, \Psi_{\mathbf{y}_j}(\mathbf{z}_N)], [\Psi_{\mathbf{y}_j}(\mathbf{z}_1^* - s\nu), \Psi_{\mathbf{y}_j}u(\mathbf{z}_2^* - s\nu), \ldots, \Psi_{\mathbf{y}_j}u(\mathbf{z}_M^* - s\nu)] \right) \right\}_{j=1}^{K}.$$

4. We define an appropriate neural network, with input size $N$ and output size $M$.
5. In the case of Problem 2, using the second approach, we used the predictions to compute the approximation in (4). Additionally, in both approaches, we applied a post-smoothing.

In all cases, we first tried to use a simple ANN with only one input and one output layer, which are densely connected. The activation function was originally a simple linear one, and no bias term was applied. Then, we tried also more layers including convolutional ones to reduce the number of parameters. For the same reason, we also tried to add a dropout layer in the original setup.

**3. Results**

In the numerical experiments, we defined a non-convex domain $\Omega$ with sharp corners to demonstrate that the method also works for non-trivial geometries. This is shown in Figure 1 together with the points and outward normal vectors, which were used in the computations:

- $\{\mathbf{z}_j\}_{j=1}^{N} \subset \partial\Omega$: red stars on the boundary, where Dirichlet boundary conditions are given:
- $\mathbf{z}_j^* = \mathbf{z}_j$: we approximated the Neumann boundary data in the same points;
- $\{\nu(\mathbf{z}_j)\}_{j=1}^{N}$: red–green–blue outward normal vectors at the boundary;
- $\{\mathbf{y}_j\}_{j=1}^{K} \subset \mathbb{R} \setminus \overline{\Omega}$: blue points (quite densely;.
- Here, according to the numerical experiments, $N = M = 89$ and $K = 10,125$.

We dealt only with Problem 2, but within the second approach, we have to compute pointwise values in (4), which is the issue of Problem 1.

Problem 2, first approach: Using $z_j = z_j^*$ with $N = M = 89$ means that the input and the output layers both have a size of 89. Assuming a dense connection between them without any intermediate layers and bias terms, we have $89 \cdot 89 = 7921$ as the number of parameters. In this way, we really need a number of learning data for a reliable fit. This suggests trying $K = 10,125$ as an optimistic guess. The forthcoming figures all correspond to this case. For an experimental error analysis, in the next subsection, we perform a series of further experiments using a variety of parameters.

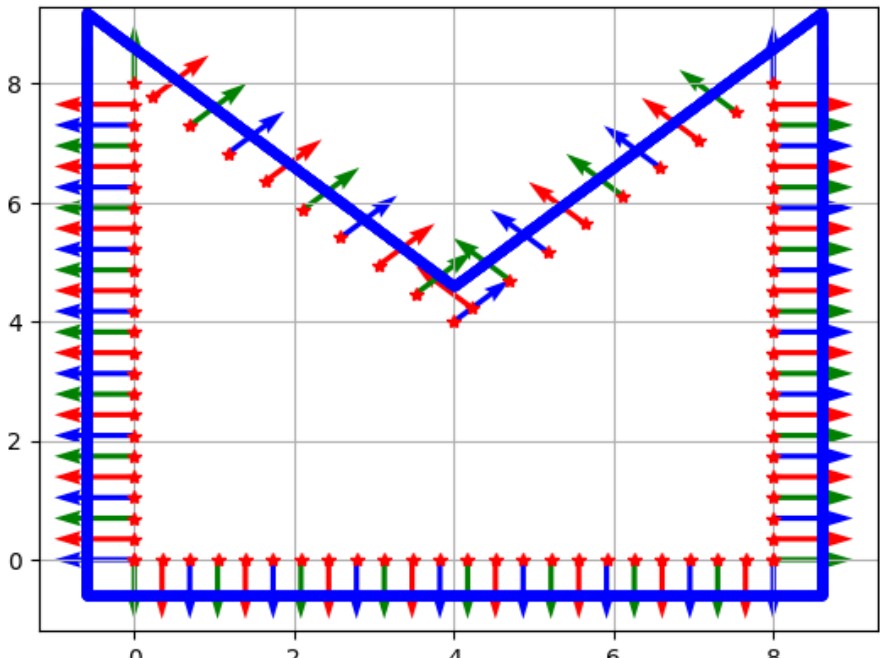

**Figure 1.** The computational domain with the boundary points (red stars), the corresponding outward normals, and the outer points (blue line).

We summarize the common setup of the corresponding computations as follows:

- Layers:
  - Full-rank approach: one input and one output layer of size $N = M$;
  - Low-rank approach: one extra intermediate layer of size 15–25% of $N$;
- Connectivity: dense, no bias term;
- Activation: linear;
- Loss function: mean-squared error;
- Optimizer: ADAM (see [16]), learning rate: 0.001, batch size: 32.

The learning performance of the neural network is shown in Figure 2. The above setup with all of the parameters was the result of a series of experiments.

To reduce the computational costs, we have more options. For this, in any case, we should reduce the number of parameters in the neural network. The simplest option is to apply an additional dropout procedure. This, however, really harms the accuracy of the approximation. We can guess this: for solving the Dirichlet-to-Neumann problems, we need dense matrices, where we cannot just change some elements to be zero. Instead, a more sophisticated strategy is to try to use more sparse layers instead of the original one dense layer. As a realistic choice, one could use, e.g., some locally connected ones. At the matrix level, this means that the original dense matrix can be well approximated with the product of some sparse ones. To mimic this, we incorporated 1, 2, 3, 4, and 5 locally connected hidden layers with kernel sizes 3, 4, and 5. This strategy, however, still did not lead to a better performance: one try is shown in Figure 3.

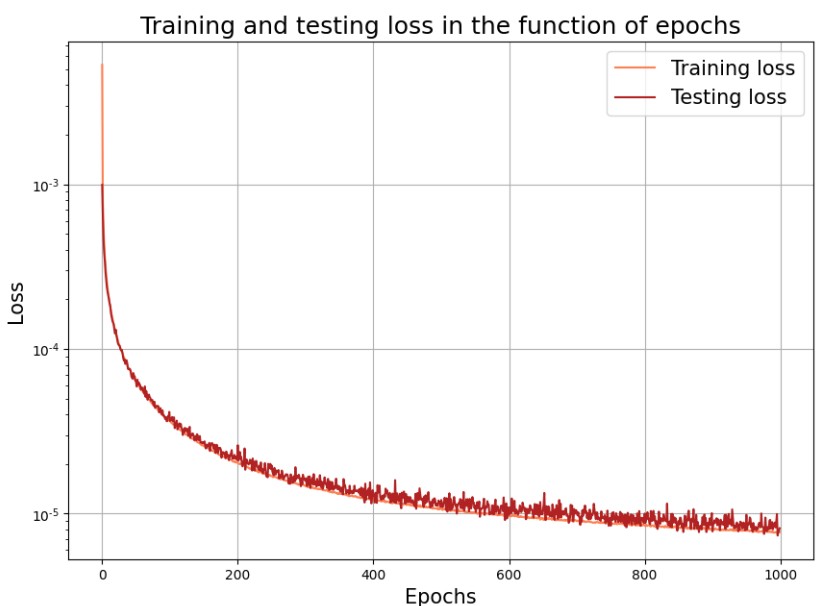

**Figure 2.** Learning performance with the fully connected input and output layers.

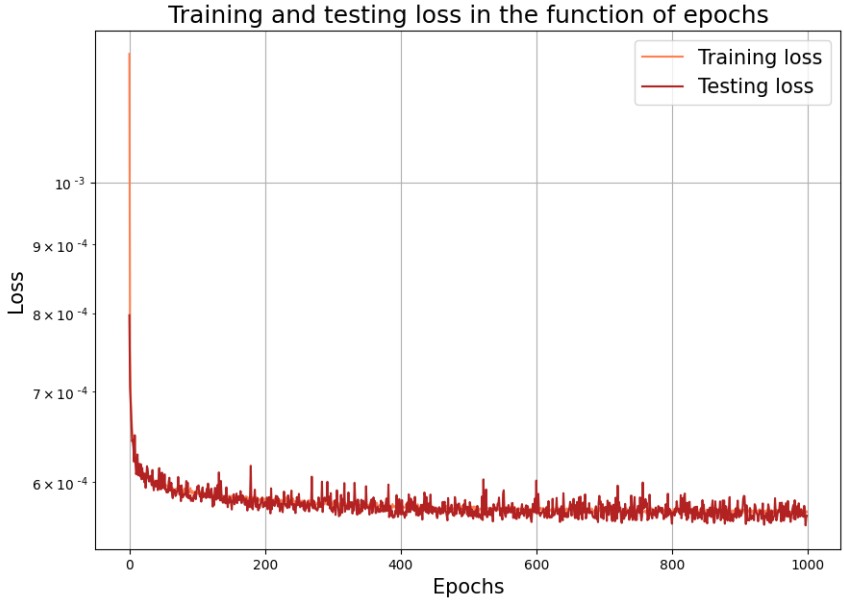

**Figure 3.** Learning performance using locally connected layers.

Therefore, in the forthcoming simulations, we use a densely connected input–output layer pair. At the end, we return back to this problem and try another strategy to reduce the number of parameters.

In the course of the second approach, using $z_j = z_j^*$ $(j = 1, 2, \ldots, N = M)$, we have to predict only the point values $u(z_j - s \cdot v)$ $(j = 1, 2, \ldots, N)$ according to the third point of our algorithm. We used the same setup as given above with $s = 0.15$, and the corresponding learning performance is depicted in Figure 4. This prediction is quite accurate, as suggested by the low loss value here.

To test the performance of the prediction, we took a model problem with the analytic solution $u : \Omega \to \mathbb{R}$, $u(x, y) = (x - 1)^2 - (y + 3)^2$. The trained neural network was applied for the boundary values of this function. We first confirmed the accuracy of the predicted point values corresponding to Figure 4. The results, which are shown in Figure 5, are convincing.

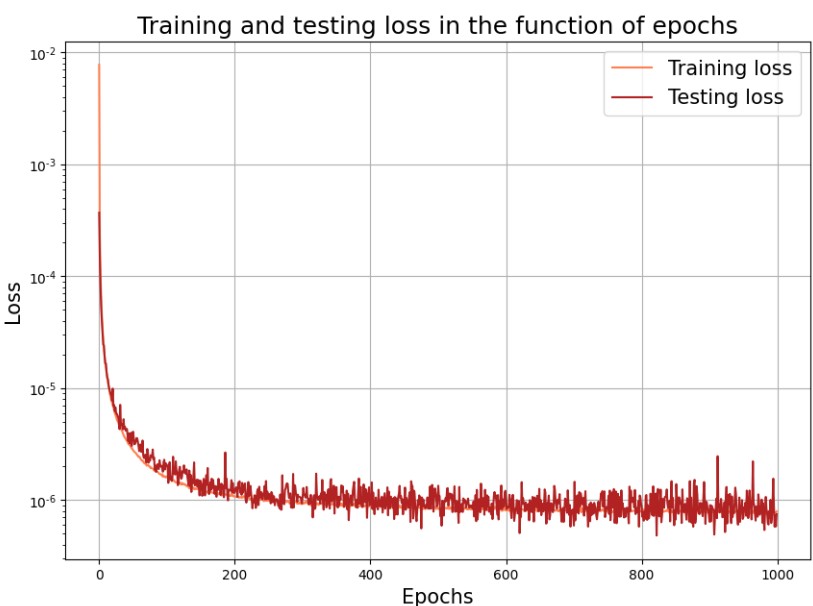

**Figure 4.** Learning performance for the interior points.

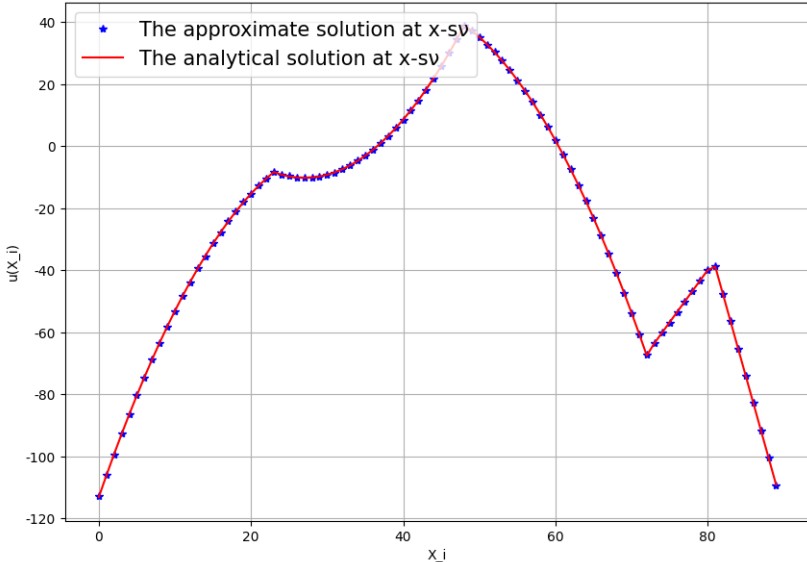

**Figure 5.** Approximation in the interior points for the computation of Neumann data.

For this test problem, we also performed a comparison between the predicted Neumann data and the real ones given with $(x, y) \rightarrow \nu(x, y) \cdot (2x - 2, -3y - 6)$. Since our domain has sharp non-convex corners with discontinuous outward normals, our approximation still needs some postprocessing. Accordingly, a kind of piecewise smoothing should be performed. For this purpose, in each case, we applied the Savitzky–Golay filter [17] with the polynomial degree 3.

The results in the case of the first and the second approach are depicted in Figures 6 and 7, respectively. Both the predicted raw data and the smoothed version are shown in the figures.

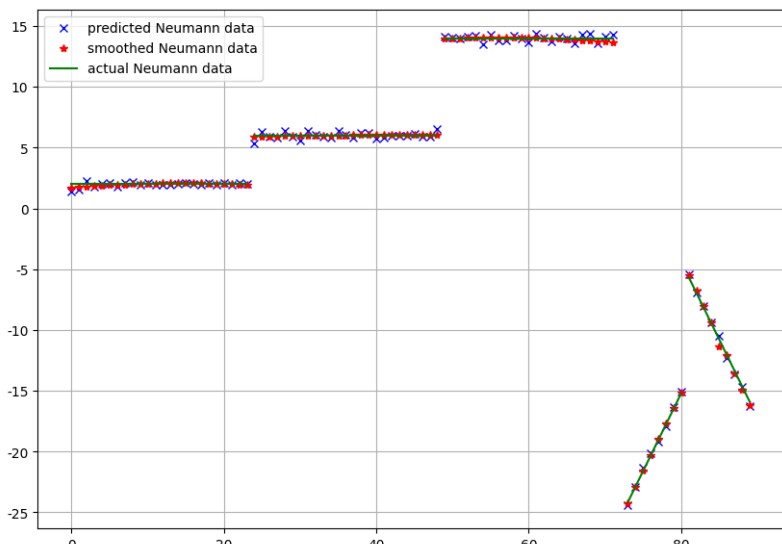

**Figure 6.** The neural network prediction of the Neumann data and the real Neumann data for the model problem in the case of the first approach.

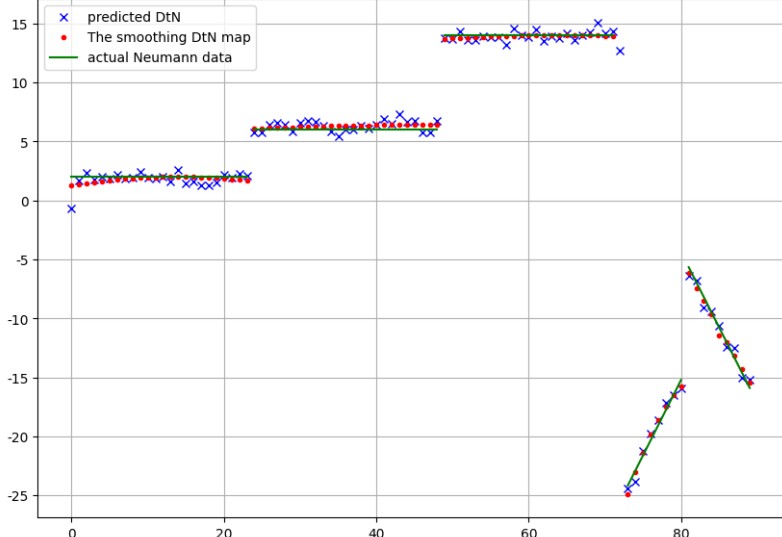

**Figure 7.** The neural network prediction of the Neumann data and the real Neumann data for the model problem in the case of the second approach.

Until now, our best neural-network-based approach has delivered a full matrix. This corresponds to a linear function transforming the Dirichlet boundary data into the Neumann data. Whenever, in practice, this really completely describes the conductivity properties of the domain $\Omega$, it contains a huge number of parameters. In the case of inverse problems, e.g., for EIT, we need these parameters as the input data. Therefore, to reduce the computational complexity of these problems, it would be highly desirable to reduce the number of parameters.

In other words, we should try to reduce the dimension of the data without a significant loss in the accuracy of our approximation of the Dirichlet-to-Neumann problem. This motivated us to perform also low-rank approximations. In practice, we can easily implement it in the framework of the neural networks. We only have to insert an extra layer of length $L$ between the input and output layers. Of course, we need a dense layer for minimizing the loss of information. In this case, the neural network corresponds to a product of matrices of size $89 \times L$ and $L \times 89$, which altogether resulted again in a matrix of size $89 \times 89$. At the

same time, whenever it is dense, its rank is only $L$ and is automatically decomposed as a product of the above matrices.

In the simulations, we tried more values $L$ to be the size of the intermediate layer. For example, taking $L = 21$ with $K = 89$ resulted in $2 \cdot 21 \cdot 89 = 3738$ parameters. Accordingly, the number of parameters was only 47 percent of the original number. Here, as we can observe in Figure 8, the learning process became faster. After some tens of epochs, the loss did not decrease any more significantly. At the same time, due to the relatively small number of parameters, we did not obtain a low loss value compared to the original model. Nevertheless, looking at the approximation of the final Neumann data in Figure 9, nearly the same accuracy was obtained as those in Figures 6 and 7.

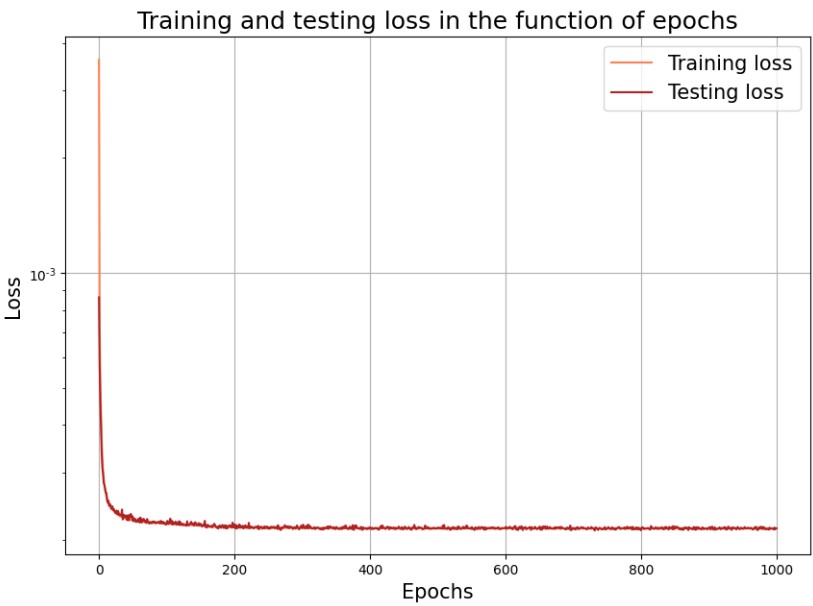

**Figure 8.** The learning performance of the neural network with one dense intermediate (hidden) layer of size 21.

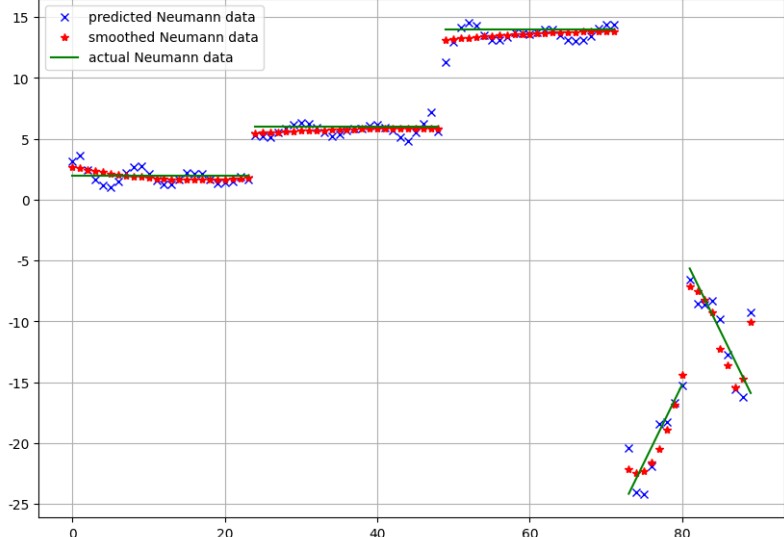

**Figure 9.** The predicted Neumann data and their smoothed version in comparison with the real one for the low-rank approximation in case of the first approach.

### 3.1. Extension to Poisson Problems

The above technique can also be extended to the case of the Poisson equation:

$$\begin{cases} \Delta u(\mathbf{x}) = f(\mathbf{x}) & \mathbf{x} \in \Omega \\ u(\mathbf{x}) = g(\mathbf{x}) & \mathbf{x} \in \partial\Omega, \end{cases} \tag{7}$$

where, in addition to (2), we have a given source function $f : \Omega \to \mathbb{R}$. We also assumed that this has an extension to a differentiable function:

$$\tilde{f} : \mathbb{R}^d \to \mathbb{R} \quad \text{with} \quad \lim_{|\mathbf{x}| \to \infty} |\mathbf{x}|^2 \cdot f(\mathbf{x}) = 0.$$

We applied here a lifting, by taking first

$$u^* = \tilde{f} * \Psi_0,$$

which, using the assumptions on $\tilde{f}$, implies

$$\Delta u^* = \Delta(\tilde{f} * \Psi_0) = \tilde{f} * \Delta\Psi_0 = \tilde{f} * \delta_0 = \tilde{f}.$$

In this way, using (7), $u - u^*$ satisfies the following Laplace equation:

$$\begin{cases} \Delta(u - u^*)(\mathbf{x}) = 0 & \mathbf{x} \in \Omega \\ (u - u^*)(\mathbf{x}) = g(\mathbf{x}) - u^*(\mathbf{x}) & \mathbf{x} \in \partial\Omega. \end{cases} \tag{8}$$

Computing an approximation of the Dirichlet-to-Neumann map for (8), we obtain $\partial_\nu(u - u^*)(\mathbf{z}_j)$. In this way, adding the known term $\partial_\nu u^*(\mathbf{z}_j)$ to this, we finally obtain the desired Neumann data $\partial_\nu u^*(\mathbf{z}_j)$.

Using Approach 1, this is given formally in the following steps.

(i)  Compute the Dirichlet data:

$$g(\mathbf{z}_j) - u^*(\mathbf{z}_j) = g(\mathbf{z}_j) - \int_{\mathbb{R}^2} \tilde{f}(\mathbf{z}_j - \mathbf{x}) \cdot \Psi_0(\mathbf{x}) \, \mathrm{d}\mathbf{x}, \; j = 1, 2, \ldots, N,$$

and similarly,

$$\partial_\nu u^*(\mathbf{z}_j) = \boldsymbol{\nu} \cdot \int_{\mathbb{R}^2} \nabla \tilde{f}(\mathbf{z}_j - \mathbf{x}) \cdot \Psi_0(\mathbf{x}) \, \mathrm{d}\mathbf{x}, \; j = 1, 2, \ldots, N.$$

(ii)  Apply the trained ANN with the input $[g(\mathbf{z}_1) - u^*(\mathbf{z}_1), \ldots, g(\mathbf{z}_N) - u^*(\mathbf{z}_N)]$.
(iii)  Add the term $[\partial_\nu u^*(\mathbf{z}_1), \ldots, \partial_\nu u^*(\mathbf{z}_N)]$ to the output of the ANN, to have an approximation of $[\partial_\nu u(\mathbf{z}_1), \ldots, \partial_\nu u(\mathbf{z}_N)]$.

### 3.2. Experimental Error Analysis

We performed a quantitative evaluation of the proposed algorithms using the above simulation results.

We begin with the experimental analysis of the first approach. The computational error between the smoothed Neumann data and the real Neumann data is displayed in a number of cases in Table 1. We applied a series of uniform discretizations of the boundary consisting of $N$ points. For each case, two different numbers $K$ of learning data were applied. This should be larger than the number of parameters, which is also shown. The first series corresponds to Figure 6, where $N = 89$ and K = 10,125. In the second series, the performance of the low-rank approximation was evaluated. Here, one intermediate dense layer was also inserted. The related number of parameters is shown in the third column. The line with K = 10,125 and $N = 89$ corresponds to Figure 9.

In the case of the second approach, we used the same parameters in the computation. At the same time, corresponding to Figure 5, we also evaluated the pointwise error in the

interior points. Recall that, here, no smoothing is necessary. In the last two columns, again, the computational error between the smoothed Neumann data and the real Neumann data is displayed. This corresponds to Figure 7. The results in the case of the low-rank approximation are also shown. See the details in Table 2.

**Table 1.** Computational error using the first approach. First half: results for the full-rank case; second half: results for the low-rank case.

| K | N = M | # of Parameters | $l_2$-Error | max-Error |
|---|---|---|---|---|
| 4500 | 61 | 3721 | 4.66 | 4.63 |
| 9000 | 61 | 3721 | 2.75 | 1.93 |
| 10,125 | 89 | 8100 | 3.29 | 2.63 |
| 20,250 | 89 | 8100 | 2.48 | 1.93 |
| 18,000 | 119 | 14,400 | 1.99 | 1.71 |
| 18,000 | 119 | 14,400 | 1.84 | 1.41 |
| 4500 | 61 | 1220 | 15.73 | 10.03 |
| 9000 | 61 | 1708 | 7.89 | 6.97 |
| 10,125 | 89 | 3738 | 5.95 | 6.24 |
| 20,250 | 89 | 4984 | 2.61 | 1.27 |
| 18,000 | 119 | 7140 | 2.26 | 1.18 |
| 18,000 | 119 | 9520 | 1.81 | 1.35 |

**Table 2.** Computational error using the second approach. First half: results for the full-rank case; second half: results for the low-rank case.

| K | N = M | # of Parameters | $l_2$-Error, Pointwise | max-Error, Pointwise | $l_2$-Error, Neumann | max-Error, Neumann |
|---|---|---|---|---|---|---|
| 4500 | 61 | 3721 | 0.97 | 0.57 | 2.58 | 1.30 |
| 9000 | 61 | 3721 | 0.94 | 0.64 | 3.24 | 1.60 |
| 10,125 | 89 | 7921 | 0.93 | 0.67 | 2.24 | 1.50 |
| 20,250 | 89 | 7921 | 1.02 | 0.60 | 2.97 | 1.45 |
| 18,000 | 119 | 14,161 | 1.06 | 0.91 | 1.433 | 0.88 |
| 18,000 | 119 | 14,161 | 1.21 | 0.55 | 2.18 | 1.04 |
| 4500 | 61 | 1220 | 6.04 | 2.65 | 14.77 | 10.38 |
| 9000 | 61 | 1708 | 3.78 | 1.57 | 6.82 | 4.48 |
| 10,125 | 89 | 3738 | 1.88 | 1.13 | 6.69 | 2.36 |
| 20,250 | 89 | 4984 | 2.13 | 1.14 | 6.29 | 3.21 |
| 18,000 | 119 | 7140 | 1.20 | 0.56 | 3.68 | 1.50 |
| 18,000 | 119 | 9520 | 1.19 | 0.50 | 3.88 | 1.83 |

## 4. Discussion and Conclusions

The mathematical basis of our approach was the approximation property of the shifted fundamental solutions $\{\Psi_{\mathbf{y}_j}\}_{j=1}^K$ of the free-space Laplacian operator. Our linear ANN-based approach can be considered as an extension of the classical MFS in the following sense:

- The ANN approach delivers an automatic way to obtain linear mappings $[g(\mathbf{z}_1), g(\mathbf{z}_1), \ldots, g(\mathbf{z}_N)] \rightarrow [u(\mathbf{x}_1), u(\mathbf{x}_2), \ldots, u(\mathbf{x}_M)]$ also with different input values. This is the case if, in an implicit discretization of time-dependent problems, more Laplacian problems have to be solved on the same domain.

We were faced with such a computational task in the implicit time discretization of problems containing the Laplacian operators:

- The optimal linear combination of the fundamental solution was found using a stochastic optimization method, which, for a large number of unknowns, can result in an efficient approach.

The ANN-based approach also confirmed that an accurate discretization of the Dirichlet-to-Neumann map should consist of a dense matrix. At the same time, using our approach, its rank can be reduced without a significant increase in the computational error.

Using the first approach, the low-rank case can deliver the same accuracy using a reduced number of parameters. The same applies to the pointwise values, which can be approximated accurately using our ANN-based approach. At the same time, the Neumann data should be approximated rather immediately using the first approach.

We finally note that, recently, a number of promising new ANN-based approaches have been developed for solving PDEs numerically. For recent reviews on this topic, see, e.g., [8,18].

**Author Contributions:** Conceptualization, F.I.; methodology, F.I.; software, T.E.D.; validation, T.E.D.; formal analysis, F.I.; investigation, F.I. and T.E.D.; resources, T.E.D.; data curation, F.I. and T.E.D.; writing—original draft preparation, F.I.; writing—review and editing, F.I. and T.E.D.; visualization, T.E.D.; supervision, F.I.; project administration, F.I.; funding acquisition, F.I. All authors have read and agreed to the published version of the manuscript.

**Funding:** This research was funded by the National Research, Development and Innovation Office within the framework of the Thematic Excellence Program 2021–National Research Sub programme: "Artificial intelligence, large networks, data security: mathematical foundation and applications".

**Data Availability Statement:** Not applicable.

**Conflicts of Interest:** The authors declare no conflict of interest.

## Abbreviations

The following abbreviations are used in this manuscript:

| | |
|---|---|
| ANN | Artificial neural network |
| EIT | Electrical impedance tomography |
| MFS | Method of fundamental solutions |
| PDE | Partial differential equation |
| PINN | Physics-informed neural network |

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
