# Peer review of "Learning Data for Neural-Network-Based Numerical Solution of PDEs: Application to Dirichlet-to-Neumann Problems"

_algorithms, doi:10.3390/a16020111_

Round 1

Reviewer 1 Report

The theme of the article is fascinating. The solution of the boundary problem for the Laplacian equation has many practical applications. Using NN for a numerical solution of a partial differential equation is a productive idea. Also, a promising concept is using the fundamental solution for NN training data.

The disadvantage of the article is missing the values of the accuracy of obtained results.  Adding numerical values of accuracy and computational errors in all cases would be better.

It is recommended to make small insignificant changes.

1. To better comprehend the Figure's title text, space should be added between a) figure 2 and  3; b) figure 4 and  5; c) figure 6 and 7; d) figure 8 and 9 titles.

2. It would be better to add some numeric presentation for accuracy and errors in calculations

3. It would be better to describe the application of the method to computational electrical impedance tomography in more detail.

4. Line 267 has a mistake of "confirms that aan accurate" that should be correct to "confirms that an accurate."

Author Response

We are grateful for the quick response and for the constructive suggestions. We have changed the text in all cases according to these. The new details are all shown in red. Many of them were suggested by the other reviewers.

  • Main critical remark: 

This kind of evaluation was really missing. We have now performed a detailed experimental error analysis. For both approaches, the error of the Neumann data, while for the second approach, also the pointwise error is shown. In both cases, we have also tested the efficiency of the low-rank approximation.

The details can be found in the new Section 3.1.

  • 1) This remark is also justified. We have inserted now a space between the figure captions in all cases for a better layout.
  • 2) See the reply for the main critical remark.
  • 3) Following the advice, we have extended the explanation with two new sentences (see lines 55-58). This is also continued in lines 94-99.
  • 4) We have corrected this typo.

Reviewer 2 Report

I think this manuscript contains an interesting work and deserves to be published. However, I think it should be significantly improved, in which my comments and questions can hopefully help.   Some quantities, abbreviations, concepts and equations are not properly defined/explained, e.g. - d in line 26. Moreover, I guess that Omega is a part of R^d, but the range of u is only R.
- the "right hand side" equation "f_j=..." in line 31
- H^(1/2) and uinH^1 in line 76 
- b_j (and the summation) in Eq. (4) Dirichlet-to-Neumann map  
- loss in Fig 2...8
- real Neumann data from line 213
Some more explanation, as well as the extension of the list of abbreviations at the end of the manuscript would be useful.
  Artificial neural networks are more frequently denoted as ANNs, 
https://en.wikipedia.org/wiki/Artificial_neural_network
so this abbreviation should be mentioned at least once to help others to find the paper.   The authors write "Accordingly, NN-based approaches were also elaborated for this, see [12,13]."
It is OK, but what is the novelty of the current manuscript compared to those papers? Are the current methods not good enough, or what is the knowledge gap which motivates the research? So there are not enough review of the previous similar works and comparison of the current manuscript to those from the point of view of novelty.

I would like to ask whether it would be useful to use other analytical solutions than the fundamental one to train the ANN? 

The original problem in Eq. (1) is well posed, but what about Problem 1 and Problem 2 which the authors want to solve? I guess that their solution is not unique.
  In Fig. 5 the approximate and the analytic solution are compared. It is nice, but the errors (the difference between them) should also been plotted. The same refers to Fig 6, 7 and 9. Otherwise, it is not really justified to state that "we obtained nearly the same accuracy was obtained as in the ones on Figure 6 and Figure 7." (Also, the repetition of the word 'obtained' should be avoided.)   There are some language and style problems in the manuscript, e.g.:
- "
a low-rank approximations are constructed" singular or plural?
- "
Keywords: boundary value probelms;" - In line 20, 'For' is redundant.
- "
the Navier–Stokes equations [7]," as well as "stochastic PDEs [8]". - line 95: "by the surface" the word 'by' is redundant. - before Eq. (4) "Keeping this in mind, any input vector ...  in the NN as a boundary data we first approximate as" I think "is first approximated as", since there cannot be two subject. - Line 175: "we both" is redundant.
- Line 237: I propose "
intermediate layer of size L=20" - Line 257: "aan" - Line 258: "Neumann map should consists a" should be "Neumann map should consist of a"

Author Response

We are very grateful for the careful and quick review and for the constructive comments. Based on these, the manuscript was revised thoroughly. The new details are shown now in red. Some of them were proposed by the other reviewers. Also, a few other misprints were found and corrected.

Moreover, I guess that Omega is a part of R^d, but the range of u is only R.

Yes, this was a serious typo, which is corrected now.

- the "right hand side" equation "f_j=..." in line 31 - H^(1/2) and uinH^1 in line 76 

We have now explained fj; see also the extension of the motivation part.

Also, the definitions of the above Sobolev spaces are included now. Additionally, the space H-1/2(Ω) is defined (see lines 74-75 and line 91).

- b_j (and the summation) in Eq. (4) - Dirichlet-to-Neumann map  

We have now explained the meaning of these coefficients (see the line after (6)).

- loss in Fig 2...8

We have extended the explanation on the physics-informed neural networks in the introduction to compare this with our appoach in concrete terms. Here, also the loss function is defined (see line30).

- real Neumann data from line 213

This is given by the analytic solution of the model problem. Now, it is given in concerete terms in line 231.

- Some more explanation, as well as the extension of the list of abbreviations at the end of the manuscript would be useful.

This is also a justified coment: we have also inculded now MFS as well.

- Artificial neural networks are more frequently denoted as ANNs, 
https://en.wikipedia.org/wiki/Artificial_neural_network
so this abbreviation should be mentioned at least once to help others to find the paper.

This is particularly useful suggestion. Accordingly, we have changed the original abbreviation everywhere.

- The authors write "Accordingly, NN-based approaches were also elaborated for this, see [12,13]."
It is OK, but what is the novelty of the current manuscript compared to those papers? Are the current methods not good enough, or what is the knowledge gap which motivates the research? So there are not enough review of the previous similar works and comparison of the current manuscript to those from the point of view of novelty.

In case of a research article this should really be the main point. In this way, we have extended the explanation of this issue. See lines 28-32 and  123-133.

In short, we also reply here: Our method can be recognized as a shortcut of the conventional physics-informed neural networks. While the original method consists of various possible right-hand sides, which has to be made zero, in our approach, it is inherently zero. This reduces the number of the parameters significantly.

- I would like to ask whether it would be useful to use other analytical solutions than the fundamental one to train the ANN? 
Yes, definitely. But there is an important theoretical issue hidden here:They have to satisfy some approximation property, since, according to (6), we want to approximate the analytic solution with the linear combination of the fundamental ones. From the theory of the MFS (see, e.g. [16] in the manuscript), we have corresponding error estimates in the present case. Probably, one can extend them for general second order elliptic equations, but I did not see any related results for Euler or Navier–Stokes equations.

-The original problem in Eq. (1) is well posed, but what about Problem 1 and Problem 2 which the authors want to solve? I guess that their solution is not unique.

Yes, they are really not well-posed. We have inserted a corresponding short remark in lines 101-102. At the same time, this is a usual real situation: we only know pointwise data. To select an „optimal” approximation, in the one usually applies some regularization technique.

- In Fig. 5 the approximate and the analytic solution are compared. It is nice, but the errors (the difference between them) should also been plotted. The same refers to Fig 6, 7 and 9. Otherwise, it is not really justified to state that "we obtained nearly the same accuracy was obtained as in the ones on Figure 6 and Figure 7."

This kind of evaluation was really missing. We have now performed a detailed experimental error analysis. For both approaches, the error of the Neumann data, while for the second approach, also the pointwise errors are shown. In both cases, we have also tested the efficiency of the low-rank approximation. We have now devoted a complete subsection (Section 3.1) to this issue. The corresponding results were summarized in Table 1 and Table 2.

- (Also, the repetition of the word 'obtained' should be avoided.)

We took care now to improve the style from this point of view.

There are some language and style problems in the manuscript, e.g.:
- "
a low-rank approximations are constructed" singular or plural?

This is singular, we have corrected accordingly.

- "Keywords: boundary value probelms;"

We have corrected it.

- In line 20, 'For' is redundant.

We have cancelled this.

- "the Navier–Stokes equations [7]," as well as "stochastic PDEs [8]".

We have corrected this term.

- line 95: "by the surface" the word 'by' is redundant.

We have cancelled this term.

- before Eq. (4) "Keeping this in mind, any input vector ...  in the NN as a boundary data we first approximate as" I think "is first approximated as", since there cannot be two subject.

Yes, this is justified. We have corrected the sentence accordingly.

- Line 175: "we both" is redundant.

We have cancelled this term.

- Line 237: I propose "intermediate layer of size L=20"

We have corrected it as suggested.

- Line 257: "aan"

We have corrected this term.

- Line 258: "Neumann map should consists a" should be "Neumann map should consist of a"

We have corrected this, too.

Reviewer 3 Report

The authors proposed an algorithm based on neural network to find the numerical solution of boundary value problems for the Laplace equation.

11.      There are some limitations in this paper. Actually, there are some analytic treatment of Laplace equation with Dirichlet and Neumann boundary conditions in literature. So, just finding numerical solution for Laplace equation is not a research problem.

22.       The obtained results are not enough to publish a research article in a repute journal.

33.      The authors can include Poison equation to get solution based on their algorithm.

44.      Overall, The paper is not suitable for publication in its present form

Author Response

 11.  There are some limitations in this paper. Actually, there are some analytic treatment of Laplace equation with Dirichlet and Neumann boundary conditions in literature. So, just finding numerical solution for Laplace equation is not a research problem.

  •   We did not simply solve Laplace equation with Dirichlet and Neumann boundary conditions. Rather, we have investigated a Dirichlet-to-Neumann problem. In the framework of our approach, for this, it is not necessary to solve a related Laplace equation. Moreover, no spatial discretization or grid generation is necessary to get the approximation.

22.  The obtained results are not enough to publish a research article in a repute journal.

  •  According to the further reviews, we tried our best to improve the manuscript. In particular, we have performed a detailed experimental error analysis to make the method and the results more convincing. Also, a number of stylistic errors were corrected.

 33.      The authors can include Poison equation to get solution based on their algorithm.

  • The Poisson equation Δ u = g  can be lifted to have a Laplace equation (with some modified boundary data). If it would be hard to find a function ug , for which   Δ ug = g, one can extend it to the whole space and the convolution g*ψ0 delivers such a function.

Reviewer 4 Report

Dear Editor,

I have reviewed the article: " Learning Data for Neural Network-Based Numerical Solution of PDEs: Application to Dirichlet-to-Neumann Problems". It is an interesting article that has merit in attracting readers. It can be considered after focusing on the following points for revision.

1). In this article nothing has been obtained for electrical impedance tomography. Therefore to avoid misleading the paper's audience the last line of the abstract "Such efficient solution algorithms 8 can serve as a basis of the computational electrical impedance tomography" may be moved to the conclusion section or better removed it.

2) Introduction section line 22, Physical informed neural networks or physically informed neural networks? Check similarly throughout

3) Introduction lines 55-57, English does not make any sense. It must be edited to make the purpose clear. Similarly check throughout the other parts of paper

4) Page 4, equation 2 is author work or it is used from some previously published work is not clear as it is stated so it must be presented more clearly by providing reference

5) Page 5 line 133,  "a rigorous and general convergence theory for the MFS is still missing". This is the actual problem to work on. Authors must state what is their contribution to achieving this aspect in this article. 

6) Page 6, line 173, the meaning is not clear. This sentence should be rewritten more explicitly to make the meaning clearer.

7) On page 8, the Label of figures does not have a good presentation. Too large that one can not focus on the figure.

8) On page 9 (Figure 6. The neural network prediction of the Neumann data and the real Neumann data for the model problem in the case of the first approach. Figure 7. The neural network prediction of the Neumann data and the real Neumann data for the model problem in the case of the second approach) Labelling should be modified

9) Page 9 line 257, is small letters aan is right or capitals should be used? Similarly, check throughout.

10) References [1] and [2] are unpublished work. Better to remove these and also these are not used for computation etc in this research. Therefore, introduction line 20-21 (For recent reviews on this topic with 20 promising new approaches can be found, e.g., in [1] and [2].) can be either removed or may be proposed in last section for future work.

11) Page 1, line 21, Probably is not a suitable word it creates doubts on the rigour of the method that the authors also stated on page 5 that rigorous of the method is missing. Therefore, I recommend that authors must focus on this aspect during the revision. 

Author Response

We are very grateful for the careful review and for the constructive comments. According to these, we tried to improve the quality of the manuscript. Also,  a number further corrections were performed. All of them are printed now with red color.

  • 1) As proposed, the last sentence was removed from the introduction.
  • 2) Indeed, this was a misprint. The correct term is "physics-informed neural network. Both in the introduction and in the abbreviation list, we have now this phrase.
  • 3) We have corrected this sentence.
  • 4) This approximation of the Neumann data involves only the definition of the directional derivative (in the outward direction). We did not see it yet to apply this to determine the Neumann data. At the same time, as the pointwise estimate of our method was rather accurate (see Figure 5 and column 4 in Table 2), we tried to use this formula. To make this point clear, we have inserted a new sentence after this formula.
  • 5) Unfortunately, this purely experimental work does not deliver any evidence for a rigorous analytic error analysis. At the same time, this practical usefulness is a further motivation to develop a full rigorous error analysis. We are working on this issue, as well.
  • 6) We have reformulated this sentence.
  • 7) We could not understand this remark. Figures on Page 8 have only a legend and a figure caption. The figure captions were really too large just that they were not separated. We have corrected this issue now at each of the figures.
  • 8) As mentioned in the previous point, we have modified the layout of the figure captions also here.
  • 9) The term "aan" was just a typo, the term "an" is the right one. We have corrected it.
  • 10) This remark is also justified. I also personally do not like to refer to preprints. At the same time, due to the rapid development of this research topic, published works do not contain the actual approaches. If I submit something, I really take care to be really up-to-date. Therefore, we have rather decided to keep these references, but as suggested, we have shifted them to the last section.
  • 11) The term "probably" was removed as suggested: we are really convinced that PINN is the most widespread ANN-based method for this purpose.  It would really be fine to set up a rigorous framework for the present approach. At the same time, this work, as the Journals title proposes was intended to set up an efficient and clear algorithm. But as I mention, parallel with this, we are also working to develop a solid basis of all of these.

Round 2

Reviewer 2 Report

The manuscript is greatly improved and I propose its acceptance.

Author Response

Dear Reviewer,

Thank you for the quick and positive response.

Reviewer 3 Report

The authors revised the paper. But still some points are not considered from the review.   They can add more good examples to support their algorithm.

Author Response

Dear Reviewer,

Thank you for the quick response.

According to the comments, we have completed the work by extending the algorithm to the case of Poisson problems. Here, one has to compute convolutions (numerically) in the boundary points, where the Dirichlet-to-Neumann map is approximated. We devoted a complete new section (Section 3.1 in the revised version) to this issue.

Also, after a new proofread, a few (four) other small corrections were performed.

Reviewer 4 Report

The authors have revised the paper according to the comments.

Author Response

(The authors gave the same response as above.)
